# SPECTRAL RETRIEVAL-AUGMENTED TIME-SERIES FORECASTING

## ABSTRACT

Time series forecasting leverages historical patterns to predict future values, but traditional methods face challenges when dealing with complex, non-stationary patterns that are difficult to memorize during training. Retrieval-augmented approaches have emerged as promising solutions by retrieving similar historical patterns to enhance predictions. However, existing retrieval methods suffer from two fundamental limitations: spectral blindness, which overlooks critical frequency-domain characteristics that capture underlying periodic structures, and temporal recency, which treats all historical data equally without emphasizing recent, more relevant patterns. In this paper, we propose SpecReTF, a novel retrieval method that addresses these issues by converting time series into windowed frequency representations, measuring similarity with a combined metric that captures both amplitude and phase information. To balance recency and historical context, we apply an exponential moving average weighting scheme that emphasizes recent windows. Extensive experiments on benchmark datasets demonstrate that SpecReTF outperforms time-domain retrieval methods, achieving superior forecasting accuracy across diverse, non-stationary time series.

## 1 INTRODUCTION

Time series forecasting is a fundamental task across numerous domains, from financial markets (Sezer et al., 2019; Mondal et al., 2014) and energy consumption (Deb et al., 2017; Koprinska et al., 2018) to economics (King, 1965; Franses, 1998) and healthcare monitoring (Zhang et al., 2024; Kaushik et al., 2020). The core challenge lies in identifying and leveraging historical patterns to predict future values, particularly when dealing with complex, non-stationary time series that exhibit irregular patterns and varying statistical properties over time. Traditional forecasting methods, including deep learning approaches, rely solely on learned representations encoded in model parameters, struggling to capture rare or complex patterns that appear infrequently in training data.

Retrieval-augmented approaches have gained prominence across machine learning, from large language models using retrieval-augmented generation (Lewis et al., 2020) to enhance factual accuracy and context understanding, to in-context reinforcement learning (Goyal et al., 2022) where retrieved demonstrations guide policy optimization. Within time series analysis, retrieval of similar historical patterns has long been a fundamental approach spanning decades of research in nearest neighbor methods, pattern matching techniques, and template-based prediction systems. Modern retrieval-augmented time series forecasting frameworks (Liu et al., 2024a; Han et al., 2025) build upon this rich foundation, incorporating recent advances in representation learning and similarity measures to enhance pattern retrieval and aggregation. These methods demonstrate significant improvements by retrieving historically relevant patterns from training data and incorporating them directly into the forecasting process. This approach reduces the learning burden on models by providing explicit access to relevant historical patterns during inference, rather than requiring complete memorization through model weights.

Existing retrieval methods, however, suffer from two key limitations—*spectral blindness*, where time-domain similarity ignores how energy is distributed across frequency bands and thus misidentifies periodic patterns, and *temporal recency*, where all past observations are weighted equally despite recent data often carrying stronger predictive power under non-stationarity. First, spectral blindness arises because retrieval using time-domain similarity metrics such as Euclidean distance,

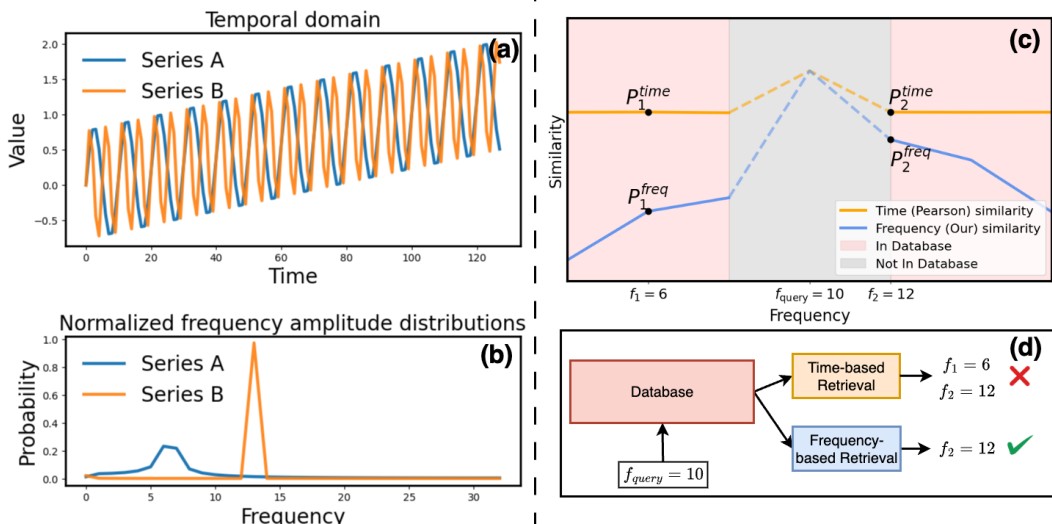

Figure 1: Limitations of Pearson correlation in capturing spectral differences. **(a)** shows the two time-series in the temporal domain. **(b)** presents their normalized frequency–amplitude distributions. **(c)** compares time-domain (Pearson) similarity (orange) and our frequency-based similarity (blue) as the target frequency varies: while Pearson correlation remains nearly constant, frequency-aware similarity correctly distinguishes relevant segments by reflecting true spectral alignment; colored regions indicate which frequencies exist in the retrieval database. **(d)** illustrates that for a query with $f_{query} = 10$, time-based similarity fails to differentiate between segments at $f_1 = 6$ and $f_2 = 12$, retrieving both with equal likelihood, whereas frequency-based retrieval accurately prioritizes and selects $f_2 = 12$.

~~Dynamic Time Warping, and~~ Pearson correlation ignores the distribution of energy across frequency bands that define periodic behaviors, making them sensitive to noise and temporal misalignments. As shown in Figure 1a–b, although two series (A and B) seem to be correlated in the temporal domain, their normalized frequency–amplitude distributions diverge significantly. When the frequency of series B is systematically varied, time-domain similarity between series A ($f_{query} = 10$) and series B remains mostly constant, failing to distinguish between candidates with different spectral content (Figure 1c: orange line). As a result, when series A searches a database without the exact matching frequency ($f = 10$), time-based retrieval may treat the available series with $f_1 = 6$ and $f_2 = 12$ as equally relevant. This illustrates its failure to capture the true underlying periodic behavior, leading to undesired retrieval (Figure 1d). Second, existing methods apply temporal uniformity, weighting all historical observations equally despite evidence that recent patterns carry greater predictive power than distant history under non-stationarity. Ignoring temporal recency may cause the retrieval model to rely excessively on outdated data, diluting the predictive signal from recent regime shifts, trends, or anomalies that are more indicative of future behavior.

Building on these insights, we propose SpecReTF, a novel retrieval-augmented time series forecasting method that performs similarity matching in the frequency domain. Our approach converts time series segments to the frequency domain using Short-time Fourier Transform (STFT) (Sejdic et al., 2009), normalizes the amplitude spectrum to create probability distributions, and computes a composite similarity score for each frame by combining Jensen–Shannon divergence for amplitude distributions with cosine similarity for phase alignment. As demonstrated in Figure 1c, our frequency amplitude-based similarity metric (blue line) accurately tracks spectral alignment, exhibiting higher scores when query and candidate frequencies align. Therefore, it correctly prioritizes the spectrally aligned candidate $f_2 = 12$ (Figure 1d), solving the limitation of *spectral blindness* by distinguishing true periodic matches that time-domain methods cannot. Moreover, to address *temporal recency*, we weight frame-level similarity scores with an exponential moving average, which boosts the influence of recent windows while assigning gradually decaying weights to older windows. This design maintains sensitivity to new patterns without forgetting persistent long-term phenomena, such as seasonal cycles and structural trends, that are critical for accurate and stable forecasting.

Our contributions can be summarized as follows:

- We propose SpecReTF, a novel retrieval-augmented forecasting architecture that combines frequency-domain analysis with recency-weighted pattern retrieval to address non-stationarity.
- We introduce a unified similarity measure that synergistically integrates Jensen–Shannon divergence on normalized amplitude spectra with cosine similarity of phase differences, effectively overcoming spectral blindness and temporal recency limitations of existing time-domain approaches.
- Through extensive evaluations on multiple benchmark datasets, we demonstrate that SpecReTF consistently achieves superior forecasting accuracy, establishing new state-of-the-art results compared to leading retrieval-based and purely model-based methods.

The remainder of this paper is organized as follows: Section 2 reviews related work in retrieval-augmented forecasting and frequency domain analysis. Section 3 presents our SpecReTF methodology in detail. Section 4 describes our experimental setup and results, and Section 5 concludes with future directions.

## 2 RELATED WORKS

### 2.1 TIME-SERIES FORECASTING

Time series forecasting has progressed from classical statistical models to advanced deep learning architectures. The Autoregressive Integrated Moving Average (ARIMA) (Nandutu et al., 2022; Mondal et al., 2014) model captures linear dependencies and accommodates non-stationarity through differencing, but is limited to modeling simple trends and seasonal patterns and cannot handle complex nonlinear dynamics or abrupt regime shifts. Transformer-based models such as iTransformer (Liu et al., 2024b) leverage self-attention to model long-range dependencies without recurrence, achieving strong performance on long-horizon forecasting tasks by dynamically focusing on relevant time points. Multiscale mixing approaches, such as TimeMixer (Wang et al., 2024a), decompose the input into hierarchical temporal representations, enabling the network to learn both local fluctuations and global trends simultaneously, which improves accuracy on datasets with multi-frequency behaviors. Cross-series relational architectures, exemplified by TimeBridge (Liu et al., 2025), capture dependencies across multiple correlated series through inter-series attention mechanisms, enhancing multivariate forecasts by leveraging shared patterns and cross-correlation structures. Frequency-domain networks such as FreTS (Yi et al., 2023) incorporate spectral normalization layers and explicit frequency-based feature extraction modules, providing robustness to non-stationarity by normalizing input features in the frequency domain and attenuating noise in unstable frequency bands. Despite these advances, purely model-based methods must internalize all necessary patterns within fixed parameter sets, limiting their adaptability. They cannot directly retrieve and leverage specific historical segments during inference, making them vulnerable to concept drift, rare events, and sudden pattern shifts that were underrepresented during training, and preventing them from exploiting localized historical contexts that could improve predictive accuracy.

### 2.2 RETRIEVAL-AUGMENTED TIME-SERIES FORECASTING

Retrieval-augmented forecasting enhances model-based approaches by equipping them with an external memory of historical series segments. RATD (Liu et al., 2024a) integrates patterns retrieved by a trained retriever into a diffusion-based generative model, allowing stochastic sampling from the retrieved contexts and improving uncertainty quantification in forecasts. RAFT (Han et al., 2025) demonstrated that retrieving the top-k most similar segments using time-domain distance metrics ~~such as Euclidean distance, Dynamic Time Warping, or~~ Pearson correlation and conditioning the model on these retrieved contexts can significantly boost accuracy. However, the similarity metrics used by both RAFT and RATD suffer from spectral blindness, as time-domain distances fail to capture periodic structures and spectral energy distributions that are often most informative for forecasting. Additionally, they exhibit temporal uniformity by weighting all retrieved segments equally, ignoring temporal recency, despite evidence that recent patterns in non-stationary series carry disproportionately greater predictive power than distant history. SpecReTF overcomes these

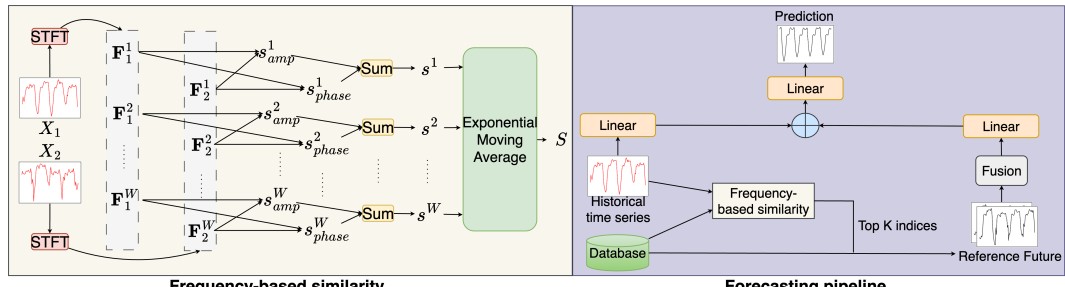

Figure 2: Overview of the SpecReTF framework. **Frequency-based similarity (left):** Retrieval mechanism applies STFT to the query and all database segments, computes frequency-based similarities using amplitude divergence and phase coherence, and selects the top-K matches via exponential recency weighting. **Forecasting pipeline (right):** The retrieved neighbors' future segments are aggregated using similarity weights, passed through a linear projection, fused with the current input history, and finally mapped to the prediction via a linear head.

limitations by employing a frequency-domain similarity measure that fuses Jensen–Shannon divergence on normalized amplitude spectra with phase cosine similarity, and by applying an exponential moving average weighting scheme to prioritize the most recent, highly relevant windows.

## 3 METHOD

### 3.1 OVERVIEW

Given a historical multivariate time series $X = \left(x_{T-L+1}, \ldots, x_T\right) \in \mathbb{R}^{L \times C}$ of length $L$ with $C$ channels, the time series forecasting problem aims to predict future values based on historical observations. Specifically, the task is to learn a function $f(\cdot)$ parameterized by $\theta$ that maps the most recent $L$ observations to the next $H$ values: $\hat{X}_{T+1:T+H} = f(X_{T-L:T})$, where $X_{T-L+1:T}$ denotes the input sequence and $\hat{X}_{T+1:T+H}$ represents the predicted future values.

In SpecReTF, we begin by treating the input series $X$ as a query window $Q = (x_{T-L+1}, \ldots, x_T)$, then compute a frequency-domain similarity score against every candidate input segment in a database $\mathcal{D}$ constructed from training samples and retrieve the top $K$ most similar pairs $\{(X_k, Y_k)\}_{k=1}^{K}$, where $X_k$ is a historical input window and $Y_k$ is its corresponding future sequence. After that, we generate two forecasts: a direct prediction from the query alone and a retrieval-based forecast computed as a weighted aggregation of the retrieved futures $\{Y_k\}$, using their similarity scores as weights.

Finally, these two outputs are merged through a fusion layer that balances the model's intrinsic forecast with evidence drawn from historical patterns, producing the final prediction $\hat{X}_{T+1:T+H}$. By combining learned representations with explicit historical context, SpecReTF enhances robustness to non-stationarity and improves the modeling of rare patterns through frequency-domain retrieval.

### 3.2 FREQUENCY-AWARE SIMILARITY

To overcome the limitations of spectral blindness and temporal recency in existing retrieval-augmented forecasting methods, we design a frequency-aware similarity measure that (1) captures both amplitude and phase characteristics in the frequency domain and (2) biases similarity toward the most recent segments via exponential weighting (Figure 2).

Given a query window $Q = \left(x_{t-L+1}, \ldots, x_t\right)$ and a candidate window $X_k = \left(x_{t'-L+1}, \ldots, x_{t'}\right)$ from the database $\mathcal{D}$, we compute their similarity score $S(Q, X_k)$ through the following steps:

**Short-Time Fourier Transform (STFT).** We partition each length-$L$ series into $W$ overlapping frames using a fixed frame size $M$ and hop size $B$. For each frame $w$, we compute the complex

STFT coefficients:

$$F_Q^w(f) \;=\; A_Q^w(f)\, e^{j\,\Phi_Q^w(f)}, \quad F_{X_k}^w(f) \;=\; A_{X_k}^w(f)\, e^{j\,\Phi_{X_k}^w(f)}, \tag{1}$$

where $f$ indexes frequency bins ($f = \{1, .., M\}$, $A(\cdot)$ denotes the amplitude spectrum, and $\Phi(\cdot)$ denotes the phase spectrum. The STFT transforms each time-domain frame into its spectral representation, making periodic structures and oscillatory behavior explicit.

**Amplitude Distribution Normalization.** To compare frequency content independent of overall signal power, we normalize each amplitude spectrum into a probability distribution:

$$p_Q^w(f) = \frac{A_Q^w(f)}{\sum_f A_Q^w(f)}, \quad p_{X_k}^w(f) = \frac{A_{X_k}^w(f)}{\sum_f A_{X_k}^w(f)}. \tag{2}$$

This normalization ensures that the similarity metric focuses on how energy is distributed across frequencies, achieving robustness to scaling differences and amplitude variations between series.

**Amplitude Similarity via Jensen–Shannon Divergence.** We quantify the difference between the two normalized amplitude distributions using the Jensen–Shannon divergence (JSD), a symmetric and bounded measure of distributional dissimilarity:

$$d_{\mathrm{JS}}^w = \mathrm{JSD}\big(p_Q^w \,\|\, p_{X_k}^w\big). \tag{3}$$

Since higher divergence indicates greater dissimilarity, we convert it into an amplitude similarity score for each frame:

$$s_{\mathrm{amp}}^w = 1 - d_{\mathrm{JS}}^w, \tag{4}$$

so that $s_{\mathrm{amp}}^w \in [0, 1]$, with 1 indicating identical amplitude distributions.

**Phase Similarity via Cosine of Mean Phase Difference.** While amplitude spectra capture the energy distribution, phase spectra encode the temporal alignment of oscillatory components. We compute the mean phase difference across frequencies for frame $w$:

$$\Delta\Phi^w = \frac{1}{M} \sum_{f=1}^{M} \big[\Phi_Q^w(f) - \Phi_{X_k}^w(f)\big], \tag{5}$$

and derive a phase coherence score using the cosine function:

$$s_{\mathrm{phase}}^w = \cos\big(\Delta\Phi^w\big), \tag{6}$$

which lies in $[-1, 1]$. A value close to 1 indicates strong alignment of phase patterns, while values near $-1$ indicate antiphase behavior.

**Frame-Level Composite Score.** For each frame $w$, we fuse amplitude and phase similarities into a single composite score:

$$s^w = s_{\mathrm{amp}}^w + s_{\mathrm{phase}}^w. \tag{7}$$

This summation balances spectral energy overlap and phase coherence, ensuring that both amplitude and temporal alignment contribute to the similarity assessment.

**Recency-Weighted Aggregation.** Recent work (Johnsen et al., 2024) on non-stationary time series has shown that recent observations often carry stronger predictive signals than older ones. To reflect this temporal recency bias, we aggregate the frame-level scores using an exponential moving average:

$$S(Q, X_k) = \sum_{w=1}^{W} \alpha \, (1 - \alpha)^{W-w} \, s^w, \quad \alpha \in (0, 1), \tag{8}$$

where $\alpha$ is a decay factor controlling how quickly the influence of older frames diminishes. Larger $\alpha$ assigns more weight to the latest frames, enabling SpecReTF to prioritize recent spectral alignments. As demonstrated in Figure 3, tuning $\alpha$ impacts forecast accuracy across different sequence lengths.

The final similarity score $S(Q, X_k)$ integrates normalized amplitude comparisons, phase coherence, and temporal recency into a single scalar measure. This comprehensive metric allows SpecReTF to retrieve historical segments that not only share underlying periodic structures and phase alignment with the query but also emphasize the most recent, and thus most predictive, patterns.

## 3.3 SPECRETF FRAMEWORK

Building on our frequency-aware similarity measure, SpecReTF integrates retrieval and forecasting into a unified end-to-end framework. Figure 2 illustrates the complete pipeline, which consists of two main components: a retrieval mechanism that identifies historically similar patterns using frequency-domain analysis, and a forecasting pipeline that aggregates retrieved patterns to produce the final prediction.

**Retrieval Mechanism.** Given a query window $Q = (x_{t-L+1}, \dots, x_t)$, the retrieval mechanism first applies STFT to transform both the query and all candidate windows in the historical database $\mathcal{D}$ into their frequency representations. For each candidate window $X_k \in \mathbb{R}^{L \times C}$, we compute the frequency-aware similarity score $S(Q, X_k)$ using the method described in Section 3.2. The retrieval system then ranks all candidate windows by their similarity scores and selects the top-$K$ most similar segments:

$$\mathcal{R}(Q) = \{(X_k, Y_k, S_k)\}_{k=1}^{K}, \tag{9}$$

where $Y_k \in \mathbb{R}^{H \times C}$ represents the future continuation of historical window $X_k$, and $S_k = S(Q, X_k)$ denotes the corresponding similarity score. This retrieval process ensures that selected patterns share both spectral characteristics and recent temporal dynamics with the query.

**Forecasting Pipeline.** The forecasting pipeline operates on both the original query and the retrieved historical patterns. The retrieved future segments $\{Y_k\}_{k=1}^{K}$ are first aggregated using their similarity scores as weights:

$$Y_{\text{retrieval}} = \frac{\sum_{k=1}^{K} \exp(S_k) Y_k}{\sum_{i=1}^{K} \exp(S_i)} \tag{10}$$

creating a similarity-weighted average of the retrieved future patterns. This aggregated retrieval forecast $Y_{\text{retrieval}}$ is then passed through a linear projection to make a retrieval-based prediction $\hat{Y}_{\text{retrieval}}$.

Simultaneously, the original query $Q$ is processed through a separate pathway to generate a direct forecast $\hat{Y}_{\text{direct}}$ using a linear layer. This direct pathway ensures that the model retains its ability to make predictions based on learned patterns even when retrieval provides limited guidance.

The retrieval-based and direct forecasts are concatenated and then mapped to the target horizon:

$$\hat{Y}_{\text{final}} = \text{Linear}(\text{Concat}(\hat{Y}_{\text{retrieval}}, \hat{Y}_{\text{direct}})), \tag{11}$$

where $\text{Concat}(\cdot)$ denotes vector concatenation.

Table 1: Comparison of **SpecReTF** and baseline methods across 8 datasets using MSE. For all datasets, results are averaged over three different seeds and forecasting horizons of 96, 192, 336, and 720. Best performances are **bolded**, and the second-best are underlined. Full results are listed in Appendix C.

| Methods | SpecReTF | RAFT | TimeMixer | PatchTST | TimesNet | MICN | DLinear | FEDformer | Stationary | Autoformer | Informer |
|---|---|---|---|---|---|---|---|---|---|---|---|
| ETTh1 | **0.415** | 0.422 | 0.448 | 0.515 | 0.493 | 0.479 | 0.463 | 0.499 | 0.571 | 0.495 | 1.049 |
| ETTh2 | **0.340** | 0.356 | 0.366 | 0.390 | 0.413 | 0.572 | 0.565 | 0.436 | 0.525 | 0.452 | 4.429 |
| ETTm1 | **0.343** | 0.349 | 0.380 | 0.411 | 0.403 | 0.422 | 0.407 | 0.445 | 0.480 | 0.588 | 0.961 |
| ETTm2 | **0.248** | 0.255 | 0.273 | 0.291 | 0.291 | 0.355 | 0.356 | 0.305 | 0.303 | 0.327 | 1.407 |
| Electricity | **0.157** | 0.162 | 0.180 | 0.217 | 0.193 | 0.195 | 0.228 | 0.212 | 0.192 | 0.227 | 0.314 |
| Exchange | 0.445 | 0.442 | 0.388 | 0.564 | 0.415 | **0.318** | 0.646 | 1.192 | 0.462 | 1.447 | 2.475 |
| Traffic | **0.431** | 0.436 | 0.485 | 0.525 | 0.622 | 0.594 | 0.627 | 0.615 | 0.620 | 0.628 | 0.763 |
| Weather | **0.238** | 0.242 | 0.241 | 0.264 | 0.255 | 0.268 | 0.262 | 0.306 | 0.289 | 0.338 | 0.631 |
| Best | **7** | 0 | 0 | 0 | 0 | 1 | 0 | 0 | 0 | 0 | 0 |

This architecture enables SpecReTF to leverage both explicit historical patterns through frequency-aware retrieval and implicit learned representations through direct forecasting, providing robustness across diverse forecasting scenarios and datasets.

# 4 EXPERIMENTS

## 4.1 EXPERIMENT SETTINGS

**Dataset.** We use eight standard datasets spanning multiple domains and time scales: ETT (hourly and 15-minute electricity transformer temperatures), Electricity (hourly household power consumption), Exchange (daily currency rates), Traffic (hourly freeway occupancy), and Weather (10-minute meteorological readings). For details of datasets, please refer to Appendix A.

**Baselines.** We compare against the retrieval-augmented method RAFT, and leading model-based forecasters: Autoformer (Wu et al., 2021), Informer (Zhou et al., 2021), FEDformer (Zhou et al., 2022), PatchTST (Nie et al., 2023), Stationary (Liu et al., 2022), DLinear (Zeng et al., 2023), TimeMixer (Wang et al., 2024b), TimesNet (Wu et al., 2023), and MICN. This array spans statistical, transformer, and lightweight architectures, enabling direct evaluation of our frequency-domain retrieval against existing retrieval strategies and contemporary forecasting models.

**Implementation details.** We conduct all experiments on a single NVIDIA Tesla V100. We take MSE (Mean Squared Error) as the loss function and MAE (Mean Absolute Error) as the evaluation metric, and the results are averaged across three different seeds and all prediction lengths. STFT uses Hanning windows with 50% overlap. Models are trained with AdamW, a batch size of 32, weight decay of 1e-4, and early stopping (with a 10-epoch patience). For additional implementation details, please refer to Appendix B.

## 4.2 BENCHMARKING RESULTS

Table 1 presents the comprehensive comparison of SpecReTF against state-of-the-art baselines across eight benchmark datasets, with results averaged over forecasting horizons. SpecReTF achieves the best performance on seven out of eight datasets and ranks second on the remaining (Exchange), demonstrating consistent superiority of spectral retrieval over traditional approaches.

**Performance against Retrieval-Augmented Methods.** SpecReTF consistently outperforms RAFT across seven of eight datasets, achieving improvements of 4.5% on ETTh2, 3.1% on Electricity, and 2.7% on ETTm2, with an average improvement of 2.0%. The only exception is Exchange, where RAFT retains a marginal 0.7% lead. The consistent advantage across diverse domains validates the robustness of our frequency-based retrieval compared to time-based counterparts.

Table 2: Ablation study of retrieval mechanism on ETTh1, ETTh2, ETTm1, ETTm2, and Weather datasets. **no retrieval** removes the retrieval modules, leaving only the linear predictor. **random retrieval** randomly retrieves relevant examples without using the similarity metric. For all datasets, results are averaged across three different seeds and forecasting horizons of 96, 192, 336, and 720. Best performances are **bolded**.

| Methods | ETTh1 | ETTh2 | ETTm1 | ETTm2 | Weather |
|---|---|---|---|---|---|
| SpecReTF | **0.415** | **0.340** | **0.343** | **0.248** | **0.238** |
| no retrieval | 0.425 | 0.351 | 0.357 | 0.260 | 0.263 |
| random retrieval | 0.423 | 0.350 | 0.359 | 0.259 | 0.267 |

Table 3: Ablation study of the similarity metric on ETTh1, ETTh2, ETTm1, ETTm2, and Weather datasets. **average temporal aggregation** replaces the recency-weighted aggregation with the average operation. **only amplitude similarity** removes the the phase similarity. **only phase similarity** removes the frequency amplitude similarity. For all datasets, results are averaged across three different seeds and forecasting horizons of 96, 192, 336, and 720. Best performances are **bolded**.

| Methods | ETTh1 | ETTh2 | ETTm1 | ETTm2 | Weather |
|---|---|---|---|---|---|
| SpecReTF | **0.415** | **0.340** | **0.343** | **0.248** | **0.238** |
| average temporal aggregation | 0.421 | 0.345 | 0.350 | 0.254 | 0.242 |
| only amplitude similarity | 0.419 | 0.342 | 0.346 | 0.250 | 0.239 |
| only phase similarity | 0.431 | 0.349 | 0.354 | 0.257 | 0.246 |

**Comparison with Model-Based Methods.** SpecReTF significantly outperforms purely model-based approaches, achieving substantial improvements over transformer architectures such as PatchTST, and TimesNet. On the challenging ETTm2 dataset, SpecReTF outperforms PatchTST by 14.5% and TimesNet by 14.8%. On Exchange, however, its dependence on past analogues offers limited benefit due to frequent pattern shifts, allowing parametric models like MICN and TimeMixer to perform slightly better. These results highlight the value of explicit pattern retrieval over purely parametric learning, especially when historical patterns are informative for future predictions.

### 4.3 ANALYSIS

**Ablation study** To quantify the benefit of our retrieval mechanism, we compare SpecReTF against two simplified baselines: a linear predictor without any retrieval modules (**no retrieval**) and a variant that retrieves historical segments at random without using our frequency-based similarity metric (**random retrieval**). Table 2 shows that across five benchmark datasets (ETTh1, ETTh2, ETTm1, ETTm2, and Weather) SpecReTF achieves the lowest average MSE, with removal of retrieval increasing error by up to 10.9% and random retrieval yielding only marginal improvements over the no-retrieval model. These results demonstrate that our retrieval method is critical for identifying and leveraging the most informative historical patterns beyond what direct prediction can achieve.

To assess the impact of each component in our similarity metric and aggregation strategy, we perform a component-wise ablation study (Table 3). Replacing recency-weighted aggregation with uniform averaging increases MSE by up to 2.5%, confirming the importance of emphasizing newer observations. Excluding phase similarity results in a 0.8%–2.2% degradation, while removing amplitude similarity causes the largest performance drop up to 3.4% on the Weather dataset, highlighting the paramount role of spectral energy distribution comparison. Overall, these results confirm that all components are essential to SpecReTF's superior forecasting performance.

**Impact of Rencency-weighted aggregation.** Figure 3 shows the mean squared error (MSE) on ETTh1 and Exchange as the decay factor $\alpha$ varies from 0 to 0.2. At $\alpha = 0$ (uniform weighting), all frames contribute equally, resulting in higher error due to dilution of recent, informative patterns by distant history. At $\alpha = 0.2$ (strong recency bias), the model overemphasizes recent frames, neglect-

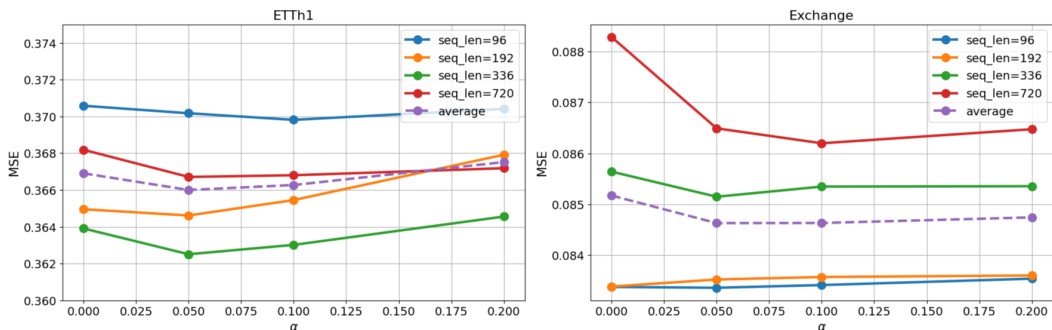

Figure 3: Impact of the decay factor $\alpha$ on forecasting performance. Solid lines show MSE for each input length and the dashed line indicates the average MSE across all lengths on **(a)** ETTh1 and **(b)** Exchange. Optimal performance occurs at intermediate $\alpha$ values, which yield the best trade-off between recency and long-term context.

Table 4: Performance comparison of PatchTST with and without the proposed retrieval module, showing average MSE across forecasting horizons. Best performances are **bolded**.

| Methods | ETTh1 | ETTh2 | ETTm1 | ETTm2 | Weather |
|---|---|---|---|---|---|
| PatchTST | 0.515 | 0.390 | 0.411 | 0.291 | 0.264 |
| PatchTST+Retrieval | 0.502 | 0.387 | 0.381 | 0.289 | 0.231 |

ing long-term context and thereby increasing error. The optimal performance occurs at $\alpha = 0.05$, which effectively balances recent and historical information, downweighting outdated frames while retaining essential stability, thereby enabling SpecReTF to adapt to evolving periodic behaviors and non-stationarity without sacrificing long-term pattern continuity.

**Generalizable Retrieval Enhancement for Forecasting Models.** We integrate our frequency-aware retrieval mechanism with PatchTST, a leading patching-based model, by directly adding the retrieval result to the output of PatchTST. As shown in Table 4, the retrieval-augmented PatchTST consistently outperforms the original architecture across all five datasets, with notable improvements of 7.3% on ETTm1 and 12.5% on Weather. These results confirm that our spectral retrieval approach is model-agnostic and can effectively boost the performance of established forecasting frameworks, demonstrating its broad applicability beyond our specific SpecReTF architecture.

**Hyperparameter Study.** We conduct a comprehensive study of key hyperparameters to understand their impact on forecasting performance. The results highlight two trade-offs. For $K$, too few segments limit context diversity while too many add noise, with the best value usually falling in the mid-range. For $M$, small windows miss spectral detail whereas large windows blur temporal changes, and intermediate sizes consistently yield the best accuracy by balancing resolution and localization. Complete results and detailed analysis are provided in Appendix D.

## 5 CONCLUSION

In this paper, we introduce a spectral retrieval-augmented time series forecasting method that addresses spectral blindness and temporal uniformity in existing approaches. By computing similarity through Jensen–Shannon divergence for amplitude distributions and cosine similarity for phase coherence, our method captures spectral characteristics overlooked by time-domain methods. To mitigate temporal recency, we aggregate frame-level similarity scores via an exponential moving average, emphasizing recent dynamics while still retaining the influence of longer-term patterns. Comprehensive experiments on eight benchmark datasets demonstrate consistent superiority, with ablation studies confirming that each component contributes meaningfully to forecasting accuracy. Future work will focus on developing automated methods for tuning frequency windowing and decay hyperparameters to improve adaptability across diverse non-stationary environments.

REPRODUCIBILITY STATEMENT

Full implementation details and experimental settings are provided in the Appendix. Following publication, we will release the source code along with comprehensive instructions to facilitate reproducibility.

LLM USAGE

Large Language Models (LLMs) were not used in the development, implementation, or evaluation of our approach. Their role was limited to improving the readability of the paper by correcting grammar and enhancing clarity of expression.

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

APPENDIX

## A  DATASET

This section provides comprehensive descriptions of the eight benchmark datasets used to evaluate SpecReTF's performance across diverse domains and temporal characteristics.

**ETT (Electricity Transformer Temperature).** The ETT dataset contains electricity transformer monitoring data from two regions in China, spanning July 2016 to July 2018. We utilize four variants: ETTh1 and ETTh2 with hourly measurements (14,400 timesteps), and ETTm1 and ETTm2 with 15-minute intervals (57,600 timesteps). Each dataset includes seven features: oil temperature (OT) and six load measurements (HUFL, HULL, MUFL, MULL, LUFL, LULL) representing high/medium/low useful and useless loads. The datasets are designed to evaluate long-sequence forecasting capabilities, particularly for energy management and equipment monitoring applications.

**Electricity.** This dataset records hourly electricity consumption in kilowatt-hours (kWh) for 321 residential and commercial clients from 2012-2014. Originally collected at 15-minute intervals, the data was aggregated to hourly resolution and filtered to remove periods with zero consumption. The dataset spans 26,304 hours with 321 variables, representing diverse consumption patterns across different customer types and usage behaviors.

**Exchange Rate.** The exchange rate dataset contains daily foreign exchange rates for eight major currencies (Australian Dollar, British Pound, Canadian Dollar, Swiss Franc, Chinese Yuan, Japanese Yen, New Zealand Dollar, Singapore Dollar) against the US Dollar from 1990-2016. With 6,071 daily observations across 8 currency pairs, this financial dataset exhibits high volatility and non-stationary behavior characteristic of foreign exchange markets.

**Traffic.** This dataset describes hourly road occupancy rates (normalized between 0 and 1) from 862 sensors deployed across San Francisco Bay Area freeways, covering January 2015 to December 2016. The California Department of Transportation (Caltrans) Performance Measurement System (PeMS) provides this high-dimensional traffic data with 17,544 hourly measurements, capturing complex spatial-temporal traffic flow patterns and congestion dynamics.

**Weather.** The weather dataset contains 21 meteorological indicators recorded every 10 minutes throughout 2020 in Germany. Variables include air temperature, humidity, wind speed, atmospheric pressure, precipitation, and solar radiation measurements from the German Weather Service (Deutscher Wetterdienst). With 52,696 10-minute observations across 21 features, this high-frequency dataset presents challenges in modeling rapid weather fluctuations and multi-scale atmospheric dynamics.

These datasets collectively span temporal resolutions from 10 minutes to daily intervals, feature dimensions from 7 to 862 variables, and application domains including energy, finance, transportation, and meteorology. The diversity in data characteristics—from smooth transformer temperatures to volatile exchange rates and complex traffic patterns—provides a comprehensive evaluation framework for time series forecasting methods.

## B  IMPLEMENTATION DETAILS

All experiments were conducted on a single NVIDIA V100 GPU. The model hyperparameters, including learning rates, batch sizes, STFT window lengths, hop sizes, number of retrieved segments K, and recency decay factor $\alpha$, are specified in Table 5. Our codebase is implemented in PyTorch 1.13 and relies on CUDA 11.7 for GPU acceleration. Complete installation instructions, environment setup guidelines (including required Python packages and version constraints), and scripts for data preprocessing, training, and evaluation are provided in the supplementary materials to facilitate full reproducibility.

Table 5: Default parameters of SpecReTF.

| Hyperparameter | Description | Choices |
|---|---|---|
| batch_size | The batch size for training | 32 |
| seq_len | Lookback window length | 720 |
| alpha | recency decay factor | {0.01 0.05 0.1 0.2} |
| train_epochs | Number of training epochs | 10 |
| patience | Early stopping patience | 5 |
| window_size | window size of STFT | {8, 16, 32} |
| learning_rate | learing rate | {0.001, 0.0001} |

## C  FULL RESULTS

We present comprehensive forecasting results for SpecReTF across 8 benchmark datasets, reporting both Mean Squared Error (Table 6) and Mean Absolute Error (Table 7) for all prediction horizons.

Table 6: Full evaluation results with MSE.

| Dataset | | SpecReTF | RAFT | TimeMixer | PatchTST | TimesNet | MICN | DLinear | FEDformer | Stationary | Autoformer | Informer |
|---|---|---|---|---|---|---|---|---|---|---|---|---|
| | | | | | | Methods | | | | | | |
| ETTh1 | 96 | 0.366 | 0.370 | 0.377 | 0.461 | 0.380 | 0.425 | 0.398 | 0.410 | 0.512 | 0.454 | 0.873 |
| | 192 | 0.400 | 0.410 | 0.428 | 0.510 | 0.432 | 0.451 | 0.446 | 0.474 | 0.535 | 0.511 | 1.016 |
| | 336 | 0.429 | 0.439 | 0.485 | 0.545 | 0.467 | 0.516 | 0.495 | 0.504 | 0.533 | 0.599 | 1.120 |
| | 720 | 0.467 | 0.470 | 0.498 | 0.544 | 0.520 | 0.524 | 0.514 | 0.608 | 0.644 | 0.527 | 1.187 |
| | Avg | 0.415 | 0.422 | 0.448 | 0.515 | 0.493 | 0.479 | 0.463 | 0.499 | 0.571 | 0.495 | 1.049 |
| ETTh2 | 96 | 0.269 | 0.276 | 0.289 | 0.308 | 0.340 | 0.372 | 0.340 | 0.357 | 0.476 | 0.346 | 3.754 |
| | 192 | 0.332 | 0.347 | 0.372 | 0.392 | 0.401 | 0.491 | 0.483 | 0.490 | 0.512 | 0.453 | 5.600 |
| | 336 | 0.366 | 0.375 | 0.386 | 0.427 | 0.452 | 0.607 | 0.590 | 0.495 | 0.552 | 0.482 | 4.720 |
| | 720 | 0.395 | 0.436 | 0.413 | 0.435 | 0.460 | 0.818 | 0.843 | 0.402 | 0.562 | 0.527 | 3.643 |
| | Avg | 0.340 | 0.356 | 0.365 | 0.390 | 0.413 | 0.572 | 0.564 | 0.436 | 0.525 | 0.452 | 4.429 |
| ETTm1 | 96 | 0.295 | 0.302 | 0.320 | 0.352 | 0.338 | 0.364 | 0.346 | 0.379 | 0.386 | 0.505 | 0.670 |
| | 192 | 0.325 | 0.329 | 0.361 | 0.390 | 0.373 | 0.402 | 0.382 | 0.426 | 0.458 | 0.553 | 0.793 |
| | 336 | 0.353 | 0.355 | 0.390 | 0.421 | 0.409 | 0.436 | 0.415 | 0.445 | 0.494 | 0.621 | 1.210 |
| | 720 | 0.401 | 0.406 | 0.449 | 0.481 | 0.491 | 0.486 | 0.485 | 0.531 | 0.582 | 0.672 | 1.171 |
| | Avg | 0.343 | 0.349 | 0.380 | 0.411 | 0.403 | 0.422 | 0.407 | 0.445 | 0.480 | 0.588 | 0.961 |
| ETTm2 | 96 | 0.162 | 0.175 | 0.183 | 0.183 | 0.186 | 0.196 | 0.201 | 0.190 | 0.255 | 0.363 | 0.431 |
| | 192 | 0.216 | 0.217 | 0.237 | 0.254 | 0.248 | 0.283 | 0.269 | 0.278 | 0.279 | 0.279 | 0.531 |
| | 336 | 0.259 | 0.275 | 0.289 | 0.309 | 0.319 | 0.379 | 0.351 | 0.352 | 0.323 | 0.338 | 1.361 |
| | 720 | 0.354 | 0.391 | 0.391 | 0.412 | 0.409 | 0.462 | 0.603 | 0.401 | 0.355 | 0.327 | 3.405 |
| | Avg | 0.248 | 0.255 | 0.273 | 0.291 | 0.291 | 0.355 | 0.356 | 0.305 | 0.303 | 0.327 | 1.407 |
| Electricity | 96 | 0.131 | 0.133 | 0.153 | 0.190 | 0.167 | 0.179 | 0.209 | 0.192 | 0.168 | 0.200 | 0.192 |
| | 192 | 0.145 | 0.149 | 0.166 | 0.199 | 0.183 | 0.188 | 0.209 | 0.201 | 0.181 | 0.222 | 0.295 |
| | 336 | 0.160 | 0.168 | 0.185 | 0.217 | 0.197 | 0.197 | 0.222 | 0.214 | 0.199 | 0.230 | 0.298 |
| | 720 | 0.192 | 0.197 | 0.216 | 0.262 | 0.224 | 0.216 | 0.272 | 0.240 | 0.220 | 0.256 | 0.371 |
| | Avg | 0.157 | 0.162 | 0.180 | 0.217 | 0.193 | 0.195 | 0.228 | 0.212 | 0.192 | 0.227 | 0.314 |
| Exchange | 96 | 0.089 | 0.091 | 0.095 | 0.084 | 0.106 | 0.101 | 0.080 | 0.147 | 0.110 | 0.196 | 0.846 |
| | 192 | 0.190 | 0.191 | 0.107 | 0.180 | 0.225 | 0.174 | 0.156 | 0.270 | 0.218 | 0.299 | 1.203 |
| | 336 | 0.392 | 0.395 | 0.349 | 0.509 | 0.366 | 0.371 | 0.304 | 0.459 | 0.491 | 0.689 | 1.671 |
| | 720 | 1.108 | 1.091 | 0.899 | 1.483 | 0.963 | 0.726 | 0.655 | 1.193 | 1.089 | 1.446 | 2.481 |
| | Avg | 0.445 | 0.442 | 0.388 | 0.564 | 0.415 | 0.318 | 0.646 | 1.192 | 0.462 | 1.447 | 2.475 |
| Traffic | 96 | 0.410 | 0.413 | 0.462 | 0.526 | 0.592 | 0.576 | 0.649 | 0.586 | 0.611 | 0.609 | 0.717 |
| | 192 | 0.427 | 0.435 | 0.473 | 0.522 | 0.616 | 0.588 | 0.597 | 0.603 | 0.612 | 0.611 | 0.694 |
| | 336 | 0.438 | 0.442 | 0.497 | 0.516 | 0.628 | 0.593 | 0.604 | 0.620 | 0.617 | 0.662 | 0.691 |
| | 720 | 0.449 | 0.454 | 0.508 | 0.537 | 0.653 | 0.619 | 0.658 | 0.651 | 0.640 | 0.631 | 0.651 |
| | Avg | 0.431 | 0.436 | 0.485 | 0.525 | 0.622 | 0.594 | 0.627 | 0.615 | 0.620 | 0.628 | 0.763 |
| Weather | 96 | 0.160 | 0.165 | 0.163 | 0.186 | 0.172 | 0.198 | 0.194 | 0.216 | 0.172 | 0.265 | 0.298 |
| | 192 | 0.213 | 0.216 | 0.220 | 0.234 | 0.233 | 0.223 | 0.231 | 0.216 | 0.199 | 0.261 | 0.368 |
| | 336 | 0.257 | 0.267 | 0.250 | 0.283 | 0.245 | 0.284 | 0.281 | 0.338 | 0.230 | 0.358 | 0.576 |
| | 720 | 0.322 | 0.320 | 0.282 | 0.326 | 0.271 | 0.366 | 0.342 | 0.454 | 0.356 | 0.366 | 0.683 |
| | Avg | 0.238 | 0.242 | 0.241 | 0.264 | 0.255 | 0.268 | 0.262 | 0.306 | 0.289 | 0.338 | 0.631 |

Table 7: Full evaluation results with MAE.

| Dataset | | Methods | | | | | | | | | | |
|---|---|---|---|---|---|---|---|---|---|---|---|---|
| | | SpecReTF | RAFT | TimeMixer | PatchTST | TimesNet | MICN | DLinear | FEDformer | Stationary | Autoformer | Informer |
| ETTh1 | 96 | 0.394 | 0.395 | 0.398 | 0.445 | 0.399 | 0.444 | 0.410 | 0.421 | 0.488 | 0.393 | 0.709 |
| | 192 | 0.417 | 0.425 | 0.419 | 0.474 | 0.427 | 0.459 | 0.439 | 0.467 | 0.502 | 0.479 | 0.788 |
| | 336 | 0.440 | 0.456 | 0.456 | 0.494 | 0.466 | 0.485 | 0.465 | 0.496 | 0.532 | 0.493 | 0.806 |
| | 720 | 0.481 | 0.476 | 0.481 | 0.515 | 0.501 | 0.527 | 0.513 | 0.544 | 0.621 | 0.517 | 0.868 |
| | Avg | 0.433 | 0.438 | 0.439 | 0.482 | 0.448 | 0.479 | 0.457 | 0.482 | 0.536 | 0.496 | 0.793 |
| ETTh2 | 96 | 0.337 | 0.343 | 0.340 | 0.349 | 0.374 | 0.372 | 0.341 | 0.395 | 0.456 | 0.386 | 1.523 |
| | 192 | 0.379 | 0.392 | 0.391 | 0.404 | 0.413 | 0.490 | 0.477 | 0.437 | 0.491 | 0.451 | 1.929 |
| | 336 | 0.414 | 0.431 | 0.413 | 0.435 | 0.451 | 0.553 | 0.539 | 0.485 | 0.549 | 0.484 | 1.833 |
| | 720 | 0.444 | 0.472 | 0.434 | 0.449 | 0.467 | 0.653 | 0.659 | 0.472 | 0.558 | 0.509 | 1.623 |
| | Avg | 0.394 | 0.410 | 0.395 | 0.409 | 0.426 | 0.517 | 0.504 | 0.447 | 0.514 | 0.458 | 1.727 |
| ETTm1 | 96 | 0.347 | 0.348 | 0.356 | 0.373 | 0.374 | 0.386 | 0.373 | 0.417 | 0.397 | 0.474 | 0.569 |
| | 192 | 0.363 | 0.366 | 0.380 | 0.392 | 0.386 | 0.407 | 0.390 | 0.440 | 0.443 | 0.495 | 0.667 |
| | 336 | 0.381 | 0.382 | 0.403 | 0.413 | 0.410 | 0.430 | 0.414 | 0.458 | 0.463 | 0.536 | 0.869 |
| | 720 | 0.408 | 0.412 | 0.413 | 0.448 | 0.448 | 0.461 | 0.450 | 0.489 | 0.515 | 0.560 | 0.821 |
| | Avg | 0.374 | 0.377 | 0.393 | 0.406 | 0.404 | 0.421 | 0.407 | 0.451 | 0.454 | 0.516 | 0.732 |
| ETTm2 | 96 | 0.223 | 0.191 | 0.200 | 0.182 | 0.205 | 0.296 | 0.286 | 0.226 | 0.273 | 0.338 | 0.451 |
| | 192 | 0.294 | 0.295 | 0.298 | 0.316 | 0.345 | 0.359 | 0.327 | 0.327 | 0.338 | 0.339 | 0.561 |
| | 336 | 0.329 | 0.328 | 0.320 | 0.339 | 0.350 | 0.428 | 0.364 | 0.362 | 0.361 | 0.371 | 0.885 |
| | 720 | 0.354 | 0.391 | 0.395 | 0.403 | 0.402 | 0.521 | 0.524 | 0.414 | 0.412 | 0.431 | 1.336 |
| | Avg | 0.300 | 0.301 | 0.319 | 0.335 | 0.326 | 0.401 | 0.375 | 0.332 | 0.346 | 0.370 | 0.808 |
| Electricity | 96 | 0.228 | 0.231 | 0.246 | 0.295 | 0.271 | 0.292 | 0.301 | 0.307 | 0.272 | 0.316 | 0.367 |
| | 192 | 0.243 | 0.246 | 0.259 | 0.303 | 0.321 | 0.301 | 0.304 | 0.314 | 0.285 | 0.342 | 0.385 |
| | 336 | 0.256 | 0.258 | 0.276 | 0.318 | 0.299 | 0.311 | 0.318 | 0.328 | 0.303 | 0.441 | 0.393 |
| | 720 | 0.291 | 0.296 | 0.309 | 0.351 | 0.319 | 0.329 | 0.349 | 0.352 | 0.320 | 0.360 | 0.438 |
| | Avg | 0.254 | 0.258 | 0.273 | 0.317 | 0.303 | 0.308 | 0.318 | 0.325 | 0.295 | 0.365 | 0.396 |
| Exchange | 96 | 0.202 | 0.208 | 0.213 | 0.202 | 0.233 | 0.234 | 0.202 | 0.277 | 0.236 | 0.322 | 0.750 |
| | 192 | 0.307 | 0.323 | 0.319 | 0.301 | 0.343 | 0.315 | 0.292 | 0.379 | 0.334 | 0.368 | 0.893 |
| | 336 | 0.451 | 0.430 | 0.426 | 0.530 | 0.447 | 0.406 | 0.413 | 0.499 | 0.473 | 0.605 | 1.033 |
| | 720 | 0.785 | 0.800 | 0.701 | 0.958 | 0.745 | 0.657 | 0.600 | 0.840 | 0.768 | 0.940 | 1.314 |
| | Avg | 0.436 | 0.440 | 0.415 | 0.498 | 0.442 | 0.403 | 0.377 | 0.499 | 0.453 | 0.539 | 0.997 |
| Traffic | 96 | 0.271 | 0.284 | 0.286 | 0.346 | 0.320 | 0.359 | 0.365 | 0.337 | 0.387 | 0.390 | 0.386 |
| | 192 | 0.273 | 0.276 | 0.295 | 0.331 | 0.335 | 0.355 | 0.372 | 0.351 | 0.339 | 0.381 | 0.377 |
| | 336 | 0.279 | 0.281 | 0.319 | 0.333 | 0.335 | 0.357 | 0.372 | 0.382 | 0.319 | 0.382 | 0.417 |
| | 720 | 0.285 | 0.296 | 0.312 | 0.351 | 0.349 | 0.360 | 0.393 | 0.381 | 0.354 | 0.407 | 0.469 |
| | Avg | 0.280 | 0.284 | 0.303 | 0.340 | 0.335 | 0.358 | 0.376 | 0.363 | 0.350 | 0.390 | 0.412 |
| Weather | 96 | 0.227 | 0.221 | 0.208 | 0.226 | 0.219 | 0.259 | 0.251 | 0.295 | 0.222 | 0.335 | 0.382 |
| | 192 | 0.264 | 0.253 | 0.249 | 0.264 | 0.260 | 0.298 | 0.288 | 0.311 | 0.266 | 0.361 | 0.542 |
| | 336 | 0.300 | 0.301 | 0.286 | 0.300 | 0.336 | 0.335 | 0.330 | 0.379 | 0.337 | 0.394 | 0.521 |
| | 720 | 0.322 | 0.350 | 0.320 | 0.357 | 0.358 | 0.387 | 0.381 | 0.427 | 0.349 | 0.421 | 0.739 |
| | Avg | 0.278 | 0.281 | 0.271 | 0.287 | 0.293 | 0.320 | 0.313 | 0.353 | 0.294 | 0.378 | 0.546 |

# D  HYPERPARAMETER STUDY

Figure 4 examines the impact of the number of retrieved segments $K$ on forecasting performance. We vary $K$ from 1 to 20 and report MSE on ETTh1, ETTh2, ETTm1, and ETTm2. When $K$ increases from 1 to 10, ETTh1 and ETTm1 show a rapid MSE reduction as the model benefits from aggregating more spectrally similar contexts, after which performance plateaus. In contrast, ETTh2 and ETTm2 exhibit a slight MSE increase at small $K$ (noisy matches), peak around $K = 10$, then modestly improve or stabilize at larger $K$. These results indicate that a moderate retrieval breadth ($K \approx 10$) optimally balances the diversity of historical patterns against the risk of diluting relevant contexts. Thus, $K$ is critical: too small $K$ limits context diversity, while too large $K$ incorporates irrelevant segments, making $K = 10$ a robust default for our datasets.

Figure 5 evaluates the impact of STFT window size on forecasting accuracy by varying the window length from 16 to 128 samples. On ETTh1, MSE steadily decreases as the window grows, achieving its lowest error at 128, since larger windows capture more complete spectral information. ETTh2 exhibits a sharp drop in MSE between window sizes 16 and 32 and then plateaus, indicating that moderate window lengths suffice to capture its dominant periodicities. For ETTm1, the best performance occurs at a window size of 32, with a slight degradation at 64 before stabilizing at 128, suggesting a trade-off between spectral resolution and temporal localization. ETTm2 follows a U-shaped trend: error rises from 16 to 32, falls to a minimum at 64, and increases again at 128, reflecting the need for intermediate window lengths to balance noise smoothing with frequency detail.

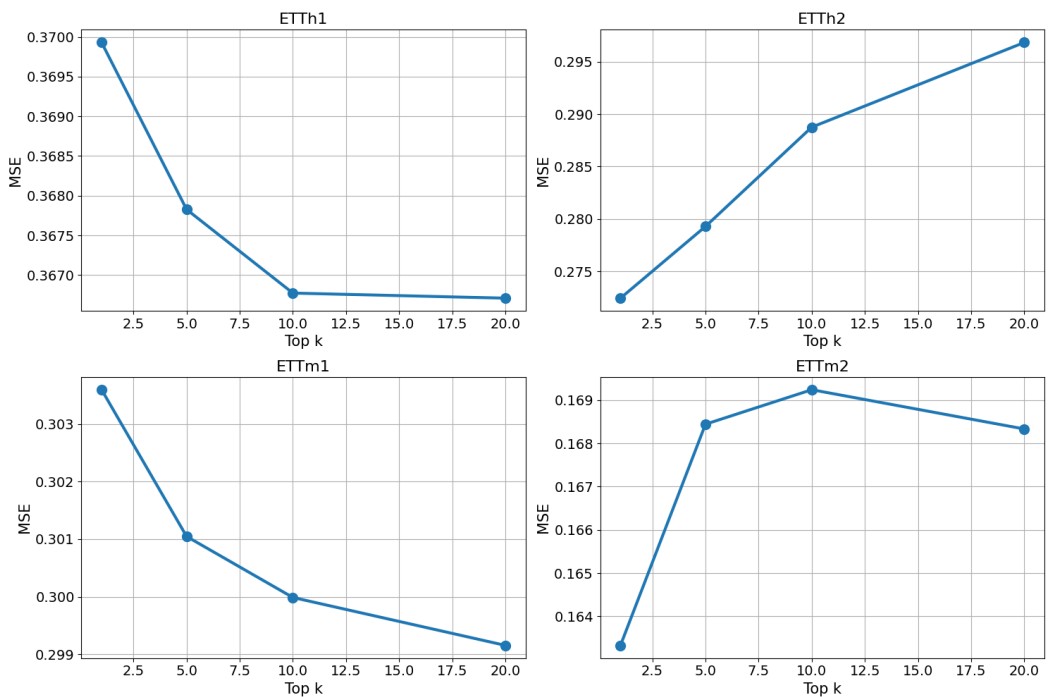

Figure 4: Analysis of the impact of the number of retrieval results.

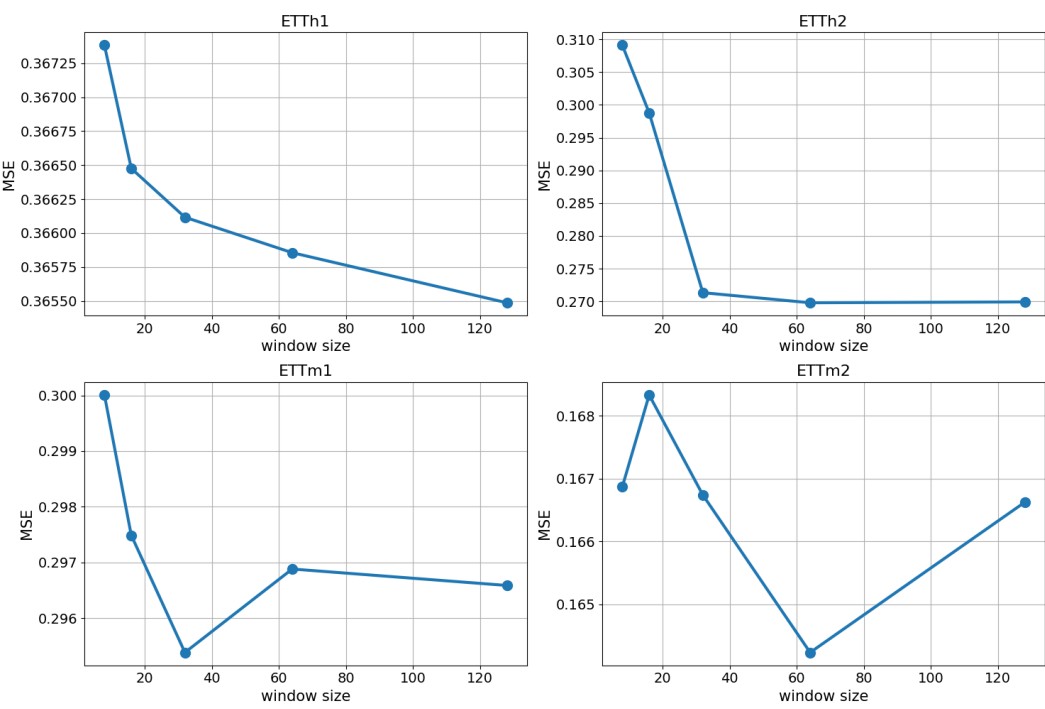

Figure 5: Analysis of the impact of the window size parameter.

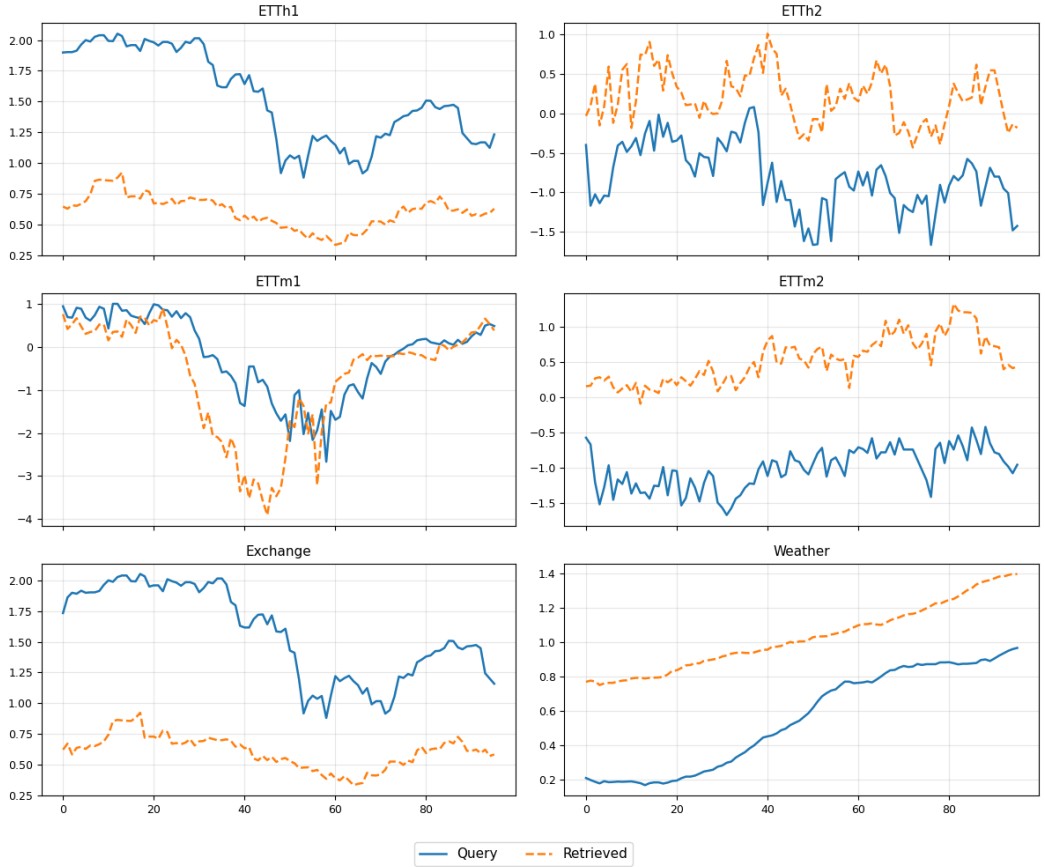

Figure 6: The example of our retrieval results on ETTh1, ETTh2, ETTm1, ETTm2, Exchange, and Weather datasets.

## E  QUALITATIVE ANALYSIS ON RETRIEVAL

Figure 6 illustrates example retrievals across six benchmark datasets (ETTh1, ETTh2, ETTm1, ETTm2, Exchange, Weather). The retrieved series closely matches the query's spectral patterns—even when temporal alignments differ—demonstrating that SpecReTF effectively identifies historically relevant contexts beyond simple time-domain similarity.

## F  COMPUTATIONAL COMPLEXITY OF THE FREQUENCY-BASED SIMILARITY METRIC

In this section, we derive the computational complexity of the proposed frequency similarity metric under the assumption of channel independence (i.e., computations are performed per-channel with trivial aggregation). We also assume a short-time Fourier transform (STFT) configuration where the hop size is approximately half the window size, i.e., $B \approx M/2$.

Consider a univariate time series of length $L$. Using an STFT with window size $M$ and hop size $B \approx M/2$, the number of resulting frames is

$$W = \left\lfloor \frac{L - M}{B} \right\rfloor + 1 \approx \frac{L}{B} \approx \frac{2L}{M}. \tag{12}$$

Each frame yields an $M$-dimensional spectrum containing the amplitude and phase values used in the similarity metric.

We now quantify the computational cost of each step.

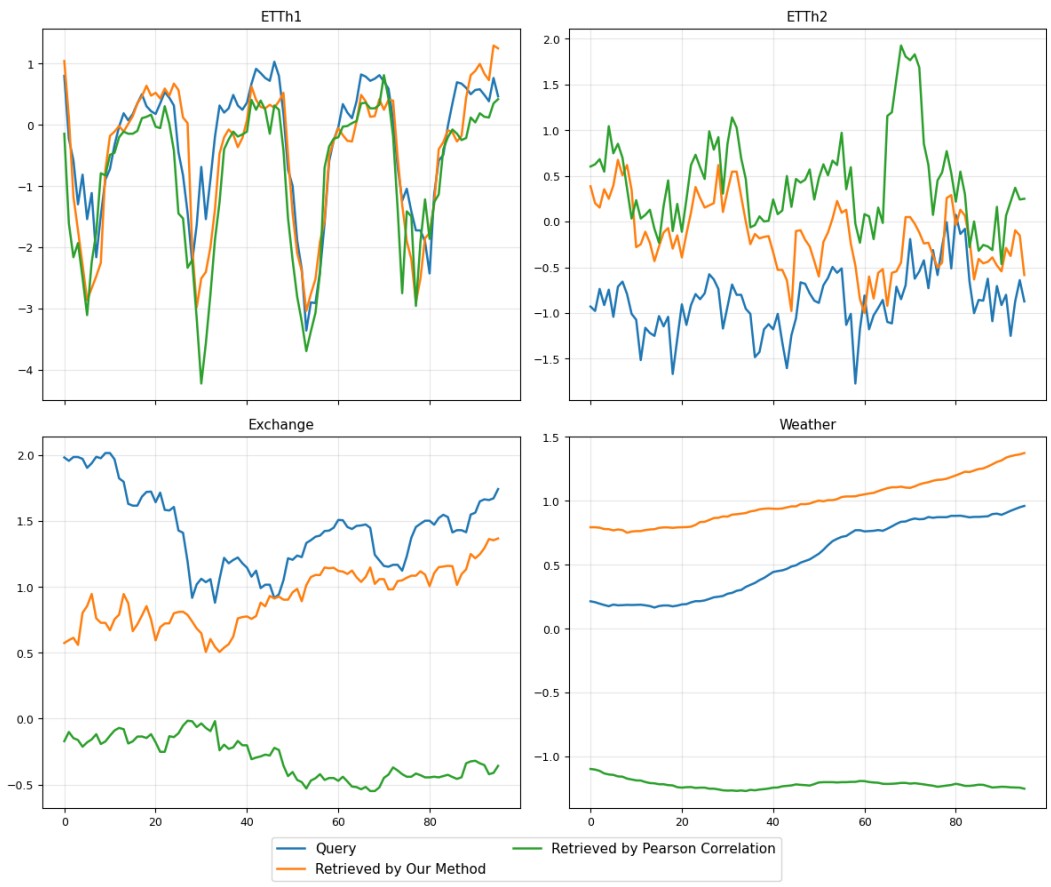

Figure 7: Comparison between retrieval results of Pearson correlation and our method on ETTh1, ETTh2, Exchange and Weather datasets.

**STFT.** Each STFT frame requires an FFT of length $M$, with cost $\mathcal{O}(M \log M)$. Over $W$ frames, the total cost is

$$T_{\text{STFT}} = \mathcal{O}(WM \log M) \approx \mathcal{O}\left(\frac{2L}{M} \cdot M \log M\right) = \mathcal{O}(L \log M). \tag{13}$$

**Amplitude Normalization.** Forming normalized amplitude distributions requires summation and rescaling of $M$ frequency bins, giving a total cost of

$$T_{\text{norm}} = \mathcal{O}(WM) \approx \mathcal{O}\left(\frac{2L}{M} \cdot M\right) = \mathcal{O}(L). \tag{14}$$

**Jensen–Shannon Divergence.** Computing the JSD involves forming the mixture distribution and evaluating two KL divergences, each costing $\mathcal{O}(M)$:

$$T_{\text{JSD}} = \mathcal{O}(WM) \approx \mathcal{O}(L). \tag{15}$$

**Phase Similarity.** The mean phase difference for each frame requires a sum across $M$ bins:

$$T_{\text{phase}} = \mathcal{O}(WM) \approx \mathcal{O}(L). \tag{16}$$

**Temporal Aggregation.** The final recency-weighted aggregation requires only $\mathcal{O}(W)$ operations:

$$T_{\text{agg}} = \mathcal{O}(W) \approx \mathcal{O}\left(\frac{L}{M}\right), \tag{17}$$

which is negligible compared to the other components.

Summing all components, the total cost of computing the similarity between a single query–candidate pair $(Q, X_k)$ is dominated by the STFT term:

$$T_{\text{total}}(Q, X_k) = \mathcal{O}(WM \log M) \approx \mathcal{O}(L \log M). \tag{18}$$

Thus, under the practically relevant configuration $B \approx M/2$, the proposed similarity metric exhibits near-linear complexity in the time-series length $L$, with only a mild logarithmic dependence on the STFT window size $M$.

