# OpenReview forum: "Spectral Retrieval-Augmented Time-Series Forecasting"
_ICLR.cc/2026/Conference — ICLR 2026 Conference Withdrawn Submission_

### Official Review · Reviewer_UYLi · 2025-10-27

**Soundness:** 3
**Presentation:** 3
**Contribution:** 3
**Rating:** 6
**Confidence:** 3

**Summary:**

This paper proposes a novel frequency-domain-based similarity method, SpecReTF, for Retrieval-Augmented time series forecasting. The work aims to address two identified limitations in existing approaches: "spectral blindness" and "temporal recency". The motivation is intuitive, and the experimental setup is relatively comprehensive.

**Strengths:**

- The paper is clearly structured, making the proposed methodology easy to follow.
- The identification of "spectral blindness" and "temporal recency" provides an easily understandable motivation for the work.
- The results demonstrate the effectiveness of the proposed method against the chosen baselines.

**Weaknesses:**

1. The two primary solutions, while effective, appear somewhat heuristic and lack elegance or a learned approach. Specifically:
- In Equation (7), the amplitude and phase similarities are combined through a simple, unweighted summation. This approach may be suboptimal, especially given that these two similarity measures are on different scales and capture distinct aspects of the signal. The work would significantly benefit from exploring automated methods for adaptively tuning or weighting the contribution of amplitude versus phase similarity.
- Similarly, the Recency-Weighted Aggregation employs a relatively rigid weighting scheme. Investigating more data-driven weighting mechanisms could be a valuable direction for future work.
2. The paper would be strengthened by a more profound discussion contrasting frequency-domain matching with established time-domain methods. Key questions remain unexplored:
3. What is the unique advantage and necessity of frequency-domain features compared to time-domain shape features (e.g., DTW) or deep learning-based embeddings? Under what scenarios would frequency-domain matching be uniquely superior?
4. What would be the effect of a hybrid approach that combines both time-domain and frequency-domain information?

**Questions:**

See Weakness.

---

> ### Author Response · Authors · 2025-11-20
> **Response to Reviewer UYLi (1/2)**
>
> We appreciate your review and the positive comments regarding our paper. We would like to respond to your comments as follows. If you find our rebuttal satisfactory, we kindly ask you to consider raising your score.
>
>
> - **In Equation (7), the amplitude and phase similarities are combined through a simple, unweighted summation. This approach may be suboptimal, especially given that these two similarity measures are on different scales and capture distinct aspects of the signal. The work would significantly benefit from exploring automated methods for adaptively tuning or weighting the contribution of amplitude versus phase similarity.**:
>
> In our current design, we intentionally apply a simple, unweighted sum to preserve interpretability and avoid overfitting, especially given the limited size of some datasets. To mitigate scale mismatch, we normalize the Jensen–Shannon divergence and cosine similarity terms to the [0, 1] range before combining them. This ensures that neither component dominates due to raw magnitude differences.
>
> Empirically, our ablation study in Appendix D shows that removing either term (“w/o amplitude” or “w/o phase”) leads to clear performance degradation across datasets, suggesting that both components are necessary and complementary, even when combined equally. While we did not explore learnable or adaptive weighting in this version, we agree that this is a promising direction for further improving retrieval quality.
>
>
>
> - **Similarly, the Recency-Weighted Aggregation employs a relatively rigid weighting scheme. Investigating more data-driven weighting mechanisms could be a valuable direction for future work.**:
>
>
> In this work, we adopt a simple exponential decay function to weight retrieved segments based on temporal recency. This design is intentionally lightweight and motivated by our core hypothesis that in non-stationary settings, recent segments are typically more informative for forecasting than older ones. The exponential decay has the advantage of being interpretable, stable, and requiring minimal tuning, thereby avoiding overfitting and ensuring broad applicability across datasets.
>
> Besides, we agree that more flexible, data-driven weighting mechanisms (e.g., attention-based weighting or learnable decay rates) could adapt better to dataset-specific dynamics. We see this as an exciting direction for future work, and potentially complementary to our frequency-based retrieval.

---

> > ### Author Response · Authors · 2025-11-20
> > **Response to Reviewer UYLi (2/2)**
> >
> > - **What is the unique advantage and necessity of frequency-domain features compared to time-domain shape features (e.g., DTW) or deep learning-based embeddings?**
> >
> > Time-domain similarity metrics such as Pearson correlation are widely used but suffer from a fundamental limitation: they ignore how signal energy is distributed across frequencies. As shown in Figure 1 of our paper, Pearson fails to distinguish between sequences that differ significantly in frequency content yet exhibit similar shapes due to alignment or phase shifts. Our method overcomes this by explicitly comparing amplitude and phase in the frequency domain, allowing for the retrieval of segments that are truly periodically aligned with the query. We would like to clarify that while Pearson correlation is a similarity measure, Euclidean distance and Dynamic Time Warping (DTW) are distance metrics and thus cannot be directly interpreted in that way. We mistakenly referred to Dynamic Time Warping (DTW) as a similarity metric. This has been corrected in the revised version of the paper.
> >
> > While deep learning-based embeddings are powerful, they often rely on the model’s ability to internalize all relevant dynamics into fixed parameters, which introduces two limitations. First, deep models may overfit to dominant or frequently seen patterns during training, especially in low-data or high-variance regimes, leading to degraded performance on underrepresented or shifting distributions. Second, since these models must encode all possible variations in their weights, they struggle to handle rare events, abrupt regime shifts, or unseen periodicities without explicit exposure during training. To validate this, we compared SpecReTF to RATD [1] (Retrieval-Augmented Time-series Diffusion), a deep retrieval baseline that uses a TCN encoder to learn embeddings for similarity search. As shown in the table below, SpecReTF achieves significantly better performance across datasets like Electricity, outperforming both RATD and the time-domain retrieval method RAFT. These results highlight the benefits of frequency-based retrieval compared to learned embeddings.
> >
> >
> > | **Method** | **Electricity (MSE/MAE)** | **Weather (MSE/MAE)** |
> > |-----------|-----------------------------|-------------------------|
> > | **SpecReFT** | **0.157/0.254** | **0.238/0.278** |
> > | RAFT | 0.162/0.258 | 0.242/0.281 |
> > | RATD | 0.190/0.279 | 0.280/0.296 |
> >
> > [1] Liu, Jingwei, et al. "Retrieval-augmented diffusion models for time series forecasting." Advances in Neural Information Processing Systems 37 (2024): 2766-2786.
> >
> >
> > - **Under what scenarios would frequency-domain matching be uniquely superior?**
> >
> > Frequency-domain matching is uniquely superior in cases where the time series contains strong periodic or oscillatory components that vary in frequency across time. In such cases, time-domain similarity metrics suffer from spectral blindness because they overlook differences in how energy is distributed across frequencies and may retrieve segments with similar shapes but mismatched periodic content. Our method addresses this by explicitly representing each input segment in the frequency domain and comparing normalized amplitude and phase information. As shown in Figure 1, this enables accurate retrieval when the input’s future dynamics are tied to its frequency characteristics, such as in non-stationary signals with shifting dominant frequencies, where time-domain methods retrieve irrelevant segments due to superficial temporal similarity.
> >
> > - **What would be the effect of a hybrid approach that combines both time-domain and frequency-domain information?**
> >
> > We agree that combining time-domain and frequency-domain information could be beneficial, particularly in cases where both local shape and periodic structure contribute to forecasting performance. While our current approach focuses on overcoming spectral blindness by prioritizing frequency-based retrieval, extending it to a hybrid framework could enhance flexibility across a wider range of time series characteristics. We consider this a promising direction for future work.

---

> > > ### Comment · Reviewer_UYLi · 2025-11-20
> > > **Response to the Rebuttal**
> > >
> > > I keep my original rating unchanged.

---

### Official Review · Reviewer_jmvm · 2025-10-30

**Soundness:** 2
**Presentation:** 2
**Contribution:** 2
**Rating:** 4
**Confidence:** 3

**Summary:**

This paper proposes the SpecReTF method to address the shortcomings of existing RAG methods in time series prediction, which neglect key frequency domain features of potential periodic structures and fail to emphasize capturing recent, more relevant patterns in time. The pipeline (i) converts times series into windowed frequency representations using Short-time Transform(STFT), (ii) uses a combined metric that incorporates amplitude and phase information to measure similarity, (iii) apply an exponential moving average weighting scheme that emphasizes recent windows. Extensive experiments on benchmark datasets demonstrate that SpecReTF outperforms time-domain retrieval methods, achieving superior forecasting accuracy across diverse, non-stationary time series.

**Strengths:**

（i）This work focuses on the practical and important issue of the ignoring the distribution of energy across frequency bands leads to the misidentification of periodic patterns and temporal relevance.

（ii）This paper proposes a novel retrieval-augmented time series forecasrting method that performs similarity matching in frequency domain, which integrates similarity score for each frame is calculated by combining Jensen-Shannon divergence (to measure amplitude distribution) and cosine similarity (to measure phase alignment). An exponential moving average is then used to weight the frame-level similarity scores, thereby enhancing the influence of the most recent window while gradually reducing the weight of older windows.

(iii) SpecReTF is a novel retrieval-augmented forecasting architecture that combines frequency-domain analysis with recency-weighted pattern retrieval to address non-stationarity.

**Weaknesses:**

(i) It lacks hyperparameters (step size, embedding dimension, etc.) for automatic selection or robustness analysis of the Short Time Fourier Transform (STFT).

(ii) Calculated a composite similarity score for each frame using only Jensen-Shannon divergence (to measure amplitude distribution) and cosine similarity (to measure phase alignment), without comparing it with other methods to explain its advantages.

(iii) The computational complexity and actual inference overhead are not detailed, but frequency domain retrieval/STFT will significantly increase the cost.

(iv) Although frequency domain similarity is emphasized, no real-world examples are shown to illustrate that the retrieved segments are indeed more reasonable.

(v) The lack of comparative experiments with other time series prediction methods that also employ retrieval-augmented demonstrates the effectiveness of the proposed method.

**Questions:**

See Weaknesses

---

> ### Author Response · Authors · 2025-11-20
> **Response to Reviewer jmvm (1/2)**
>
> We appreciate your review and the positive comments regarding our paper. We would like to respond to your comments as follows. If you find our rebuttal satisfactory, we kindly ask you to consider raising your score.
>
>
> - **W1**
>
> The choice of STFT window size affects how well the model captures periodic structures versus transient dynamics. As shown in our hyperparameter study (Appendix D, Figure 5), performance typically improves as window size increases, up to a point that too large a window may degrade accuracy due to loss of temporal precision. For example, in case the dataset is highly periodic and smooth like ETTh1, we suggest using a larger window size of 64 to improve spectral resolution. In case the dataset is noisy or changes more rapidly, like ETTm2, we suggest using a smaller window size of 16 to better preserve local temporal dynamics. As a general recommendation, we find that a window size of 32 provides a strong default across many datasets.
>
>
> Regarding the step size, we set it to half of the window size, which balances temporal resolution and computational cost. We also clarify that our method does not include any additional learnable embedding dimension for the STFT output—the raw frequency features are used directly for similarity computation in the retrieval module.
>
>
>
> - **W2**
>
> We would like to clarify that we have compared our similarity measure to Pearson correlation, both qualitatively and quantitatively:
>
> - In Figure 1, we provide an illustrative example where Pearson correlation fails to distinguish between segments with clearly different frequency characteristics, while our frequency-aware similarity measure more accurately reflects the true similarity structure.
>
> - Compared to RAFT uses Pearson correlation across multiple temporal scales, our method achieves better forecasting performance even with single-scale retrieval, as shown in our main results (Table 1).
>
> This demonstrates that our composite similarity score provides better inductive bias for retrieval in the frequency domain. Our ablation study (Appendix D) further supports our design: removing either the amplitude or phase term leads to consistent performance drops, confirming that both components are complementary and necessary.
>
> - **W3**
>
>
> To address this concern, we have added a complexity analysis in Appendix F. Specifically, we analyze the computational cost of applying STFT for similarity computation. Considering a univariate time series of length $L$, with STFT window size $M$ and hop size $B=M/2$, the overall complexity of the STFT operation is $O(LlogM)$
>  due to the use of FFT within each sliding window. This cost is efficient in practice and scales well with input length.
>
> Moreover, to assess the real-world impact on inference time, we compare SpecReTF against RAFT with single-scale and three-scale aggregation versions, which use Pearson correlation as a similarity metric, in terms of latency.
>
> | **Method**           | **Inference Time (ms)** |
> |----------------------|--------------------------|
> | SpecReFT             | 0.35                     |
> | single-scale RAFT    | 0.34                     |
> | multi-scale RAFT     | 0.45                     |
>
> SpecReFT incurs only 0.01ms additional latency over the fastest baseline, single-scale RAFT, despite using frequency-domain operations. This minimal overhead demonstrates that our method is computationally efficient in practice. Furthermore, SpecReFT is significantly faster than the multi-scale RAFT version, which is often employed to capture richer temporal dependencies but at the cost of higher latency. The efficiency of SpecReFT, which matches or beats complex multi-scale baselines, makes it highly suitable for resource-constrained forecasting scenarios.

---

> > ### Author Response · Authors · 2025-11-20
> > **Response to Reviewer jmvm (2/2)**
> >
> > - **W4**
> >
> > Figure 1 in the Introduction section serves as a concrete example, carefully constructed to illustrate the advantage of our proposed frequency-aware similarity over traditional time-domain metrics. In this figure, we present a scenario where Pearson correlation fails to distinguish between segments with distinct underlying frequencies, despite visual similarity in the time domain. As described in the text, when a query segment has frequency $f=10$ but candidate frequencies vary, Pearson correlation (orange line) remains relatively flat and insensitive to spectral mismatch. In contrast, our frequency-based similarity (blue line) peaks when frequencies align, accurately prioritizing the correct match, even when no exact frequency duplicate exists.
> >
> > We have included a real-world illustration in Figure 7 of the revised manuscript. This figure presents query and retrieved segments from the ETTh1 dataset, comparing our frequency-domain retrieval with time-domain Pearson correlation. As shown, our method retrieves segments that are spectrally aligned and structurally more consistent with the query. This real-world case further supports our claim that time-domain retrievals can be spectrally misleading, while our frequency-aware similarity better preserves meaningful alignment in forecasting contexts.
> >
> >
> > - **W5**
> >
> > Our original submission already included a detailed comparison with RAFT, a well-established retrieval-augmented time series forecasting baseline that uses the Pearson similarity metric. These results are presented in Table 1 of the paper, showing that our proposed method SpecReTF outperforms RAFT across multiple benchmark datasets. We have further extended our experimental comparisons to include RATD [1], a recent deep learning-based retrieval method that employs a TCN encoder and diffusion model. As shown in the table below, SpecReTF consistently outperforms both RAFT and RATD, validating the effectiveness of our frequency-domain similarity metric.
> >
> > | **Method** | **Electricity (MSE/MAE)** | **Weather (MSE/MAE)** |
> > |-----------|-----------------------------|-------------------------|
> > | **SpecReFT** | **0.157/0.254** | **0.238/0.278** |
> > | RAFT | 0.162/0.258 | 0.242/0.281 |
> > | RATD | 0.190/0.279 | 0.280/0.296 |
> >
> > [1] Liu, Jingwei, et al. "Retrieval-augmented diffusion models for time series forecasting." Advances in Neural Information Processing Systems 37 (2024): 2766-2786.

---

### Official Review · Reviewer_9JN7 · 2025-10-31

**Soundness:** 3
**Presentation:** 3
**Contribution:** 2
**Rating:** 4
**Confidence:** 4

**Summary:**

The paper proposes a retrieval-based time-series forecasting framework that incorporates both a spectral perspective and temporal recency into the retrieval process.

**Strengths:**

Although the work may appear somewhat incremental - mainly replacing the similarity measurement in conventional retrieval-based time-series forecasting methods with one based on the frequency domain - the authors effectively establish the motivation through Figure 1 in the introduction. The intuition behind adopting the spectral view is clearly conveyed.

**Weaknesses:**

1. In Figure 3, the model’s performance appears insensitive to the decay factor $\alpha$. This raises doubts about whether temporal recency, one of the paper’s main claimed contributions, is truly impactful. Moreover, it remains unclear how practitioners should determine an appropriate value of $\alpha$ in real-world settings.

1. In the experiments, the “No Retrieval” configuration (a simple linear predictor) in Table 2 outperforms most baselines in Table 1. This makes me question whether the experiments were conducted under a fair and consistent setting.

**Questions:**

1. Justify the necessity of temporal recency.

2. Verify the experimental setting and guarantee the fair comparison.

---

> ### Author Response · Authors · 2025-11-20
> **Response to 9JN7**
>
> We appreciate your review and the positive comments regarding our paper. We would like to respond to your comments as follows. If you find our rebuttal satisfactory, we kindly ask you to consider raising your score.
>
> - **W1+Q1**
>
>
> We respectfully clarify that temporal recency is a critical part of our similarity metric, and its impact is evident in both ablation results and hyperparameter analysis. While Figure 3 shows that forecasting performance is relatively stable across a range of decay values $\alpha$, the optimal performance clearly occurs at intermediate $\alpha = 0.05$. At $\alpha = 0$ (average aggregation), performance worsens due to over-reliance on outdated patterns. At $\alpha = 0.2$, performance degrades again due to overemphasis on the most recent frames, ignoring valuable long-term trends. To guide practitioners, we recommend using a default decay factor of $\alpha=0.05$, which consistently performs well across datasets in our experiments. This value strikes a balance between recent and long-term patterns. For domains with rapid changes, a slightly higher $\alpha$ may be preferable, while more stationary domains may benefit from smaller values.
>
>
> - **W2+Q2**
>
> We clarify that all models in Table 1 and Table 2, including SpecReTF, ablations (such as “No Retrieval”), and all baselines were trained and evaluated under the exact same hyperparameter tuning and experiment setting, following RAFT. As for the “No Retrieval” variant in Table 2, it uses the SpecReTF framework with the retrieval module removed and replaced by a linear projection layer, trained on the same input and target sequences as SpecReTF. The good performance of this variant on certain datasets (e.g., ETTh1, ETTm1) likely reflects the benefits of our design choices (e.g., strong normalization, direct forecasting head) even without retrieval. However, this model does not outperform baselines across the board, on datasets like Weather, methods such as MICN and FEDformer still achieve better results, highlighting the added value of retrieval.

---

> > ### Comment · Reviewer_9JN7 · 2025-11-20
> >
> > W1 + Q1: I still find it difficult to identify a meaningful performance improvement attributable to selecting an appropriate $\alpha$, and thus to the usefulness of temporal recency itself. The observed performance change appears to be quite small - on the order of 0.001. This raises a conceptual and empirical question for me: can temporal recency truly be considered a substantive contribution given such limited impact?
> >
> > W2 + Q2: I appreciate the clarification that all experiments were conducted under consistent settings. However, several other reviewers also raised similar concerns. Could you elaborate further on why the simple ablation of your proposed model (“no retrieval”) achieves performance that is better than or comparable to other linear baselines such as DLinear? In your response, you mentioned strong normalization and a direct forecasting head, but it is not immediately clear how these design choices lead to such improvements. In fact, the gains attributed to these design components appear to be larger than those resulting from temporal recency, which is presented as one of the main contributions of the paper.

---

> > > ### Author Response · Authors · 2025-11-23
> > > **Additional Response to Reviewer 9JN7**
> > >
> > > **W1 + Q1**
> > >
> > > We respectfully clarify that Figure 3 only analyzes the decay factor α at a fixed prediction length (96), and is not meant to represent the overall gain from temporal recency. The actual retrieval improvements become more prominent at longer prediction lengths, where models must rely more heavily on relevant historical analogs rather than short-term extrapolation. This is reflected in Table 1 and the full results (Appendix C, Table 6), where SpecReTF consistently improves over RAFT and other baselines across horizons up to 720. Further, in Table 3, removing recency weighting degrades performance across datasets, confirming that emphasizing recent patterns helps the retrieval mechanism focus on the most predictive matches, especially under non-stationarity.
> > >
> > > **W2+Q2**
> > >
> > > We followed all experimental settings, hyperparameters, and implementation details from the RAFT paper, to ensure fair comparison. The “no retrieval” baseline in our study directly corresponds to the RAFT backbone (a linear head without retrieval), and it shows competitive performance compared to other model-based baselines. However, the main focus of our work is to improve retrieval-augmented forecasting by addressing spectral blindness in prior methods like RAFT. Our proposed frequency-based similarity metric captures both amplitude and phase information in the frequency domain and, combined with temporal recency weighting, leads to consistent improvements over RAFT across 7 of 8 datasets (Table 1). These gains demonstrate the benefit of more informed retrieval, not just architectural differences.

---

> > > > ### Comment · Reviewer_9JN7 · 2025-11-26
> > > >
> > > > Thank you for the rebuttal. I have adjusted my score.

---

### Official Review · Reviewer_jqrN · 2025-11-01

**Soundness:** 2
**Presentation:** 3
**Contribution:** 2
**Rating:** 4
**Confidence:** 4

**Summary:**

This paper introduces SpecReTF, a spectral retrieval-augmented time series forecasting method that overcomes key limitations in existing approaches by decomposing time series into frequency-domain components, measuring similarity using a combined amplitude and phase metric, and incorporating temporal recency weighting. The proposed method demonstrates superior forecasting accuracy across multiple benchmark datasets, establishing new state-of-the-art performance.

**Strengths:**

1.	The paper effectively identifies and analyzes "spectral blindness" as a key limitation in existing time-domain retrieval methods, providing compelling evidence through both theoretical discussion supported by visual examples.
2.	The combination of amplitude spectrum analysis (Jensen-Shannon divergence) and phase coherence measurement (cosine similarity) effectively addresses the identified spectral blindness problem.
3.	Extensive experiments across eight diverse benchmark datasets demonstrate consistent improvements over state-of-the-art baselines, with the method achieving superior performance on most evaluation metrics.

**Weaknesses:**

1.	The "no retrieval" results in Table 2 (which uses only two linear layers) are significantly better than the DLinear results in Table 1 on several datasets. This is contradictory because DLinear, a purpose-built linear model, should perform at least as well as this simple baseline. This inconsistency raises concerns about whether all baselines in Table 1 were evaluated under the same experimental settings as SpecReTF. Please clarify this discrepancy to ensure the fairness and validity of the comparisons.

2.	The paper claims three contributions, but they are largely overlapping and describe a single core innovation: the frequency-aware similarity metric. The architectural framework and the recency-weighting scheme, while useful, appear to be supporting components rather than distinct conceptual advances. This conflation could be seen as overstate the breadth of methodological contribution. The work's primary novelty resides in the new similarity measure, and the contributions should be reframed to more accurately reflect this.

3.	The hyperparameter study in Appendix D reveals that the model's performance is sensitive to the number of retrieved segments (K) and the STFT window size, and that the optimal values for these parameters vary across different datasets.  This observed sensitivity appears to be at odds with the paper's claim of the method's robustness.

**Questions:**

1.	Why does the simple "no retrieval" linear model in Table 2 outperform the purpose-built DLinear in Table 1? Does this indicate inconsistent experimental settings for the baseline models?

2.	The three contributions essentially revolve around the frequency-domain similarity measure. Could the main contribution be reframed to focus on this core innovation?

3.	Given the sensitivity to K and STFT window size shown in Appendix D, how does this align with the paper's claim of robustness? Furthermore, how should these hyperparameters be determined in practice for new datasets?

---

> ### Author Response · Authors · 2025-11-20
> **Response to Reviewer jqrN**
>
> We appreciate your review and the positive comments regarding our paper. We would like to respond to your comments as follows. If you find our rebuttal satisfactory, we kindly ask you to consider raising your score.
>
>
> - **W1+Q1**
>
> The “no retrieval” variant in our ablation study is designed to isolate the contribution of the retrieval module by removing it and replacing it with an additional MLP forecasting head. While this variant is still relatively lightweight, it differs from DLinear in two important ways. First, DLinear uses a single linear layer to model trend and seasonal components separately, whereas our MLPs operate on the original input and learns a direct mapping without decomposing the series. Second, we apply a per-variable normalization step, where each input sequence is subtracted by the most recent value (following RAFT), which improves performance on non-stationary data. As a result, this “no retrieval” baseline may perform particularly well on datasets like ETT, which contain strong local patterns that can be captured effectively by a small MLP. We emphasize that all baselines in Table 1, including DLinear, were re-run under identical experimental settings to ensure fairness in comparison.
>
>
> - **W2+Q2**
>
> While the temporal weighting mechanism itself is simple, it is an essential part of the method that directly addresses the core limitation of temporal uniformity in retrieval. This mechanism is tightly integrated into the similarity computation and has a demonstrable impact, as shown in our ablation study (Table 3, Figure 3), where removing recency weighting leads to consistent performance degradation.
>
> Thus, even though the weighting mechanism is conceptually straightforward, we believe it constitutes a meaningful contribution because it solves a fundamental modeling limitation that has not been adequately addressed in existing retrieval-based approaches.
>
>
> - **W3+Q3**
>
>
> We respectfully clarify that our claim of robustness is made in relative terms, specifically, robustness to non-stationarity. Unlike parametric models that must internalize all temporal patterns in their weights, our retrieval-based approach explicitly accesses relevant historical segments at inference time. This allows SpecReTF to dynamically adapt to periodic shifts, especially when such patterns are underrepresented during training.
>
> It is well known in the retrieval literature that performance depends on the number of retrieved items. A smaller K may limit context diversity, while a larger K can introduce irrelevant or noisy matches. This trade-off is not unique to our method because it also appears in text retrieval, image retrieval, and prior retrieval-augmented forecasting models (RAFT). In our study (Appendix D, Figure 4), we observe that this sensitivity is manageable. For example, on cleaner and more periodic datasets like ETTh1 and ETTm1, performance improves with more retrieved neighbors, and K = 10 offers a strong default setting. In contrast, for noisier datasets like ETTm2, larger K values degrade performance due to the inclusion of irrelevant patterns. We suggest using K = 2 in such noisy domains, where fewer high-quality matches tend to yield better forecasts.
>
> In addition, the choice of STFT window size impacts performance by controlling how well the model captures periodic structures versus transient changes. In our hyperparameter study (Appendix D, Figure 5), we find that performance improves with increasing window size up to a point, after which it may degrade due to loss of temporal precision. Specifically, on periodic datasets like ETTh1, a larger window yields better spectral resolution and improves performance. In contrast, for datasets with faster, changing or noisier signals (ETTm2), a smaller window size of 16 better preserves local temporal dynamics and leads to improved accuracy.

---

### Note · Authors · 2026-01-13

I have read and agree with the venue's withdrawal policy on behalf of myself and my co-authors.